# Adaptive guidelines for the treatment of gonorrhea to increase the effective life span of antibiotics among men who have sex with men in the United States: A mathematical modeling study

**Reza Yaesoubi**[1]*, **Ted Cohen**[2], **Katherine Hsu**[3], **Thomas L. Gift**[4], **Harrell Chesson**[4], **Joshua A. Salomon**[5], **Yonatan H. Grad**[6]

**1** Department of Health Policy and Management, Yale School of Public Health, New Haven, Connecticut, United States of America, **2** Department of Epidemiology of Microbial Diseases, Yale School of Public Health, New Haven, Connecticut, United States of America, **3** Massachusetts Department of Public Health, Boston, Massachusetts, United States of America, **4** Division of STD Prevention, Centers for Disease Control and Prevention, Atlanta, Georgia, **5** Center for Primary Care & Outcomes Research, School of Medicine, Stanford University, Stanford, California, United States of America, **6** Department of Immunology and Infectious Diseases, Harvard T. H. Chan School of Public Health, Boston, Massachusetts, United States of America

* reza.yaesoubi@yale.edu

**Data Availability Statement:** The data underlying the results presented in the study are available

## Abstract

### Background

The rise of gonococcal antimicrobial resistance highlights the need for strategies that extend the clinically useful life span of antibiotics. Because there is limited evidence to support the current practice of switching empiric first-line antibiotic when resistance exceeds 5% in the population, our objective was to compare the impact of alternative strategies on the effective life spans of antibiotics and the overall burden of gonorrhea.

### Methods and findings

We developed and calibrated a mathematical model of gonorrhea transmission among men who have sex with men (MSM) in the United States. We calibrated the model to the estimated prevalence of gonorrhea, the rate of gonorrhea cases, and the proportion of cases presenting symptoms among MSM in the US. We used this model to project the effective life span of antibiotics and the number of gonorrhea cases expected under current and alternative surveillance strategies over a 50-year simulation period. We demonstrate that compared to the current practice, a strategy that uses quarterly (as opposed to yearly) surveillance estimates and incorporates both the estimated prevalence of resistance and the trend in the prevalence of resistance to determine treatment guidelines could extend the effective life span of antibiotics by 0.83 years. This is equivalent to successfully treating an additional 80.1 (95% uncertainty interval: [47.7, 111.9]) gonorrhea cases per 100,000 MSM population each year with the first-line antibiotics without worsening the burden of gonorrhea. If the annual number of isolates tested for drug susceptibility is doubled, this strategy

from the Gonococcal Isolate Surveillance Project (https://www.cdc.gov/std/gisp/default.htm) and Sexually Transmitted Disease Surveillance 2018 (https://www.cdc.gov/std/stats18/).

**Funding:** This work was supported by the U.S. Centers for Disease Control and Prevention (CDC), National Center for HIV, Viral Hepatitis, STD, and TB Prevention Epidemiologic and Economic Modeling Agreement (5NU38PS004644, https://www.cdc.gov/nchhstp/neema) to JAS. The Centers for Disease Control and Prevention contributed to study design and preparation of the manuscript. RY was supported by 1K01AI119603, YHG by R01 AI132606, and TC by R01 AI112438, all from the National Institute of Allergy and Infectious Diseases (https://www.niaid.nih.gov/). The National Institute of Allergy and Infectious Diseases had no role in study design, data collection and analysis, decision to publish, or preparation of the manuscript.

**Competing interests:** I have read the journal's policy and the authors of this manuscript have the following competing interests: RY, TC, JAS, and YHG received funding from the National Institute of Health and the US Centers for Disease Control and Prevention. KH is employed by Massachusetts Department of Public Health. TLG and HC are employed by the US Centers for Disease Control and Prevention.

**Abbreviations:** CDC, Centers for Disease Control and Prevention; GISP, Gonococcal Isolate Surveillance Project; MSM, men who have sex with men.

could increase the effective life span of antibiotics by 0.94 years, which is equivalent to successfully treating an additional 91.1 (54.3, 127.3) gonorrhea cases per 100,000 MSM population each year without increasing the incidence of gonorrhea. Study limitations include that our conclusions might not be generalizable to other settings because our model describes the transmission of gonorrhea among the US MSM population, and, to better capture uncertainty in the characteristics of current and future antibiotics, we chose to model hypothetical drugs with characteristics similar to the antibiotics commonly used in gonorrhea treatment.

## Conclusions

Our results suggest that use of data from surveillance programs could be expanded to prolong the clinical effectiveness of antibiotics without increasing the burden of the disease. This highlights the importance of maintaining effective surveillance systems and the engagement of policy makers to turn surveillance findings into timely and effective decisions.

## Author summary

### Why was this study done?

- Gonorrhea is the second most common notifiable disease in the United States and has developed resistance to all first-line antibiotics.

- The selection of antibiotics used for gonorrhea treatment is almost always empiric and based on guideline recommendations.

- There is limited evidence to support the current practice of switching the first-line antibiotic after resistance to it exceeds 5% in annual surveillance estimates.

- Our objective was to project how alternative strategies to inform the first-line treatment recommendations impact the life span of antibiotics and the overall burden of gonorrhea.

### What did the researchers do and find?

- We developed a mathematical model that describes the key characteristics of gonorrhea transmission among men who have sex with men (MSM) in the United States.

- Our model estimates the life span of antibiotics and the incidence of gonorrhea under current and alternative strategies for changing first-line empiric antibiotic treatment.

- We found that compared to the current practice, a strategy that 1) uses quarterly surveillance estimates and 2) incorporates both the estimated prevalence of resistance and the trend in the prevalence of resistance to determine treatment guidelines could extend the effective life span of antibiotics without worsening the burden of gonorrhea.

**What do these findings mean?**

- This work suggests an opportunity to optimize the use of surveillance systems to slow the spread of antibiotic-resistant strains and control the burden of gonorrhea.

- This requires enhancing the surveillance systems (e.g., by allowing for more frequent reporting of estimates and a larger number of observations) and the engagement of policy makers to turn surveillance findings into timely decisions.

- Further studies are needed to investigate the generalizability of these conclusions.

## Introduction

Gonorrhea remains a globally significant sexually transmitted infection (550,000 reported cases in 2017 in the United States [1] and an estimated 87 million cases worldwide in 2016 [2]), and the recent descriptions of resistance to standard treatments has raised concern about the global emergence of untreatable infections [3,4]. The threat of spread of untreatable gonococcal infections highlights the need for strategies to maximize the life span of existing antibiotics while providing effective treatment for infected individuals.

The selection of antibiotics used for gonorrhea treatment is almost always empiric and based on guideline recommendations because the diagnosis is usually made by nucleic acid amplification test, which does not inform on antibiotic susceptibility [5–7]. Even when culture is available, patients likely receive first-line empiric antibiotic treatment while awaiting drug-susceptibility results. In the US, current treatment guidelines are based on the prevalence of antimicrobial resistance estimated by the Gonococcal Isolate Surveillance Project (GISP) [8], a sentinel surveillance system that monitors trends in antimicrobial susceptibilities of gonococcal strains in the US [9].

Once the point estimate for prevalence of resistance to the first-line antibiotic exceeds 5% [8,10], the WHO guideline recommends switching to another antibiotic for empiric treatment [10]. However, there is limited evidence to support this 5% threshold. Increasing the threshold may extend the life span of second-line antibiotics by minimizing the use of these agents but at the cost of decreasing the probability that any given individual with gonorrhea receives effective first-line therapy. This could be associated with greater individual morbidity and may also lead to longer durations of infectiousness, facilitating further transmission of gonorrhea. In contrast, decreasing the switching threshold may increase the probability that each individual receives effective first-line therapy but also would lead to earlier and more extensive use of second-line regimens, which would be expected to shorten their life span. Beyond the cross-sectional resistance proportion, other easily observed features of resistance emergence, such as tempo of change, could also be considered in designing optimal switching policy. A rapid rise in resistance proportion, e.g., might prompt an earlier switch in recommended antibiotics than a slow increase [11].

In this study, we used a transmission dynamic model to compare the performance of different decision rules that could inform the recommendations for the first-line therapy of gonococcal infections. Specifically, we considered whether the current switching strategy based on the 5% threshold from annually reported surveillance efforts is outperformed by policies that i) use different thresholds for the percentage of isolates that are resistant, ii) incorporate

information on the trend in the percentage of isolates that are resistant, and iii) increase the frequency and/or size of drug resistance surveys.

## Methods

### Treatment of gonococcal infections

We considered a scenario in which 3 antibiotic drugs (drug A, drug B, and drug M) are available for treatment of gonorrhea. Drug A represents first-line therapy, such as ceftriaxone or azithromycin [12], and drug B represents an alternative antibiotic that may be suitable for the first-line treatment of gonorrhea, such as zoliflodacin [13] or gepotidacin [14], both of which have been over 95% effective against urogenital gonococcal infections in phase 2 trials. Drug M represents the last-line antibiotic for gonorrhea.

We assumed that drug B would be initially reserved for treatment of cases that fail treatment with drug A. The selective pressure for resistance to drug A increases as more cases of gonorrhea are treated with this drug. Following the current strategy [8,10], one would remove drug A from clinical use and replace it with drug B once a specific threshold for resistance to drug A is exceeded. Subsequently, those who fail first-line treatment with drug B will be retreated with drug M. Likewise, when the prevalence of resistance to drug B reaches a predefined threshold, drug B will be removed from the first-line therapy, and drug M will be used for both first-line and second-line therapy.

### Adaptive guidelines to inform first-line treatment recommendations

An efficient strategy to guide the first-line treatment recommendations strikes a balance between the need to maximize the effective lives of drugs A and B with the goal of treating gonococcal infections with the most effective drug available. An adaptive guideline identifies the first-line therapy drug based on cumulative observations on the resistance characteristics of the ongoing gonorrhea epidemic. In this study, we compared the performance of 4 types of adaptive guidelines in terms of their ability to prolong the effective life of drugs A and B and to prevent gonorrhea (Table 1).

The strategies "Threshold—annual" and "Threshold—quarterly" represent the guidelines that recommend switching to a new first-line drug once the resistance prevalence passes a certain threshold (e.g., 5%) [8,10]. They differ in how frequently the estimates of resistance prevalence are obtained and treatment recommendations are updated. The strategy "Threshold—annual" with a value of 5% represents the current practice because the estimates of resistance prevalence from surveillance systems (such as GISP in the US) become available on a yearly

**Table 1. Adaptive guidelines to inform first-line treatment recommendations for gonorrhea.**

| Strategies | Frequency of Decision-Making | Annual Number of Tests for Resistance | Epidemiolocal Estimates Used for Decision-Making | Policy Examples |
|---|---|---|---|---|
| Threshold—annual | Annually | 5,000 | Estimate of resistance prevalence | Switch to a new first-line drug when the point estimate of the proportion of resistant isolates exceeds $\tau$%. |
| Threshold—quarterly | Quarterly | 5,000 | Same as Threshold | Same as Threshold. |
| Threshold + trend | Annually | 5,000 | Estimate of resistance prevalence and change in the estimate of resistance prevalence | Switch to a new first-line drug when point estimate of the proportion of resistant isolates exceeds $\tau$% or the change in the estimate of resistance since the last decision point exceeds $\theta$ percentage point. |
| Enhanced threshold + trend | Quarterly | 10,000 | Same as "Threshold + trend" | Same as "Threshold + trend." |

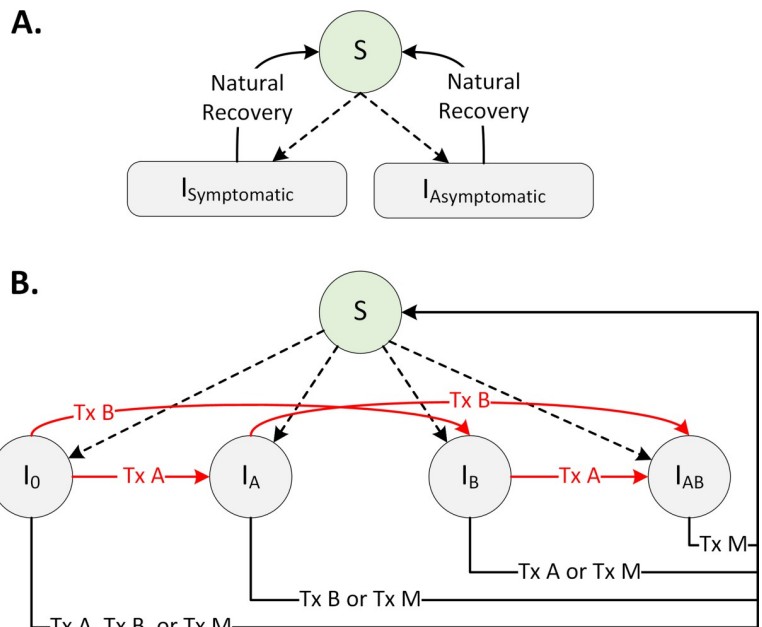

**Fig 1. A stochastic gonorrhea transmission model.** Dotted arrows represent new infection, and red arrows represent resistance acquisition while under treatment. S represents susceptibles, $I_0$ represents drug-susceptible infections, and $I_A$, $I_B$, and $I_{AB}$ represent infections resistant to drug A, drug B, and both. Tx A, Tx B, and Tx M denote treatment with drugs A, B, and M. The expanded model structure is displayed in S1 Fig in S1 Text. The model is adapted from [15], with additional details necessary to evaluate the strategies in Table 1.

basis. The "Threshold—quarterly" policy relies on the same annual number of susceptibility tests as in "Threshold—annual," but it distributes them over 4 quarters. Therefore, it might be able to detect trends in resistance more quickly but at the expense of lowering the precision in the estimates of resistance prevalence.

The strategy "Threshold + trend" seeks to detect the emergence of resistance to the first-line drug more proactively by using both estimates of resistance prevalence and the change in the resistance prevalence since the last year. An example of this strategy may recommend switching to a new first-line drug when point estimate of the proportion of resistant isolates exceeds 5% or the change in the estimate of resistance since the last decision point exceeds 1 percentage point. By accounting for the temporal change in resistance, this strategy is more responsive to the rapid spread of resistant gonococcal infection.

The strategy "Enhanced threshold + trend" is the same as strategy "Threshold + trend," except that the evaluation of resistance prevalence is performed quarterly, with twice as many annual susceptibility tests as in the strategy "Threshold + trend." Compared to the "Threshold + trend" strategy, the "Enhanced threshold + trend" strategy benefits from more frequent and a larger number of observations, which might facilitate the detection of statistically significant trends.

## A gonorrhea transmission dynamic model

To evaluate the impact of these strategies on the overall burden of gonorrhea and antibiotic life spans, we developed a stochastic compartmental model that describes the transmission of *Neisseria gonorrhoeae* among men who have sex with men (MSM) in the US (Fig 1). About 42% of gonorrhea cases in 2017 were among MSM, and the emergence of resistance among this population is of particular concern [1,5]. The model is adapted from Tuite and colleagues [15], with additional details necessary to evaluate the strategies described in Table 1.

In our model, susceptible individuals are at risk of infection with gonorrhea, and this risk varies by the prevalence of infection. Infected cases can be symptomatic or asymptomatic (Fig 1A). Infected individuals are further divided to represent the resistance profile of the infecting strain: drug-susceptible infection ($I_0$), infection resistant to drug A ($I_A$), infection resistant to drug B ($I_B$), and infection resistant to both drugs A and B ($I_{AB}$) (Fig 1B). Asymptomatic cases do not seek treatment and remain infectious until they recover spontaneously or get detected through active screening (Fig 1A). All symptomatic cases are assumed to seek treatment with some delay. Cases who seek treatment or are detected through screening will receive treatment with drug A, B, or M, depending on the current recommendation for the first-line therapy. If treated with an antibiotic to which the infecting strain is susceptible, the individual returns to the susceptible state. A portion of symptomatic individuals who fail the first-line treatment (because of receiving ineffective treatment or developing resistance) will seek retreatment with some delay. As soon as effective treatment is initiated, we assume that infected individuals no longer contribute to the force of infection (because of either negligible infectiousness and/or reduced sexual activity).

Resistance may arise while an individual receives antibiotic treatment (Fig 1B). To account for the fitness cost associated with resistance, we assumed that compared to susceptible strains, resistant strains are less transmissible [15], at least initially. Data from GISP indicate that despite the decrease in the use of tetracycline, penicillin, ciprofloxacin, cefixime, ceftriaxone, and azithromycin in recent years, the prevalence of resistance to these antibiotics has been fairly stable [1]. To produce simulated trajectories that allow for this persistence despite reduced use of these antibiotics, we allow the fitness cost of resistance to gradually decrease, consistent with the idea that the fitness costs may be compensated (see S1.3 of S1 Text) [16]. Additional details about the model are provided in S1 Text.

## Model calibration and validation

We used a Bayesian approach to calibrate our model against estimates of gonorrhea prevalence, the rate of reported gonorrhea cases in 2017, and the proportion of gonorrhea cases with symptoms. This calibration approach seeks to estimate the probability distributions of unknown parameters that result in simulated trajectories with good fit to the available epidemiological data [17]. We chose prior parameter distributions based on the available data, estimates and plausible ranges extracted from the literature, and expert opinion when estimates were unavailable (see S1 Text for additional details).

## Comparing the performance of guidelines to inform first-line treatment recommendations

We compare the performance of strategies to inform the first-line treatment recommendations (Table 1) based on the number of gonorrhea cases that could be averted with respect to the status quo (the "Threshold—annual" strategy in Table 1 with 5% switch threshold) and the increase in the effective life of drugs A and B. To measure the effective life of antibiotics, we note that the consumption of drug M is inversely related to the effective life span of drugs A and B. If resistance to drugs A and B rises quickly, implying a short effective life span for these drugs, all future cases of gonorrhea will be treated with drug M. We therefore defined the effective life span of drugs A and B as the area under the curve of the annual percentage of gonorrhea cases that are successfully treated with drugs A or B over 50 years of simulation (i.e., $\sum_{t=1}^{50} \frac{N_A(t)+N_B(t)}{N_A(t)+N_B(t)+N_M(t)}$, where $N_A(t)$, $N_B(t)$, and $N_M(t)$ are the number of gonorrhea cases treated successfully with drugs A, B, or M in simulation year $t$).

If a strategy extends the effective life span of drugs A and B by $\Delta L$ years, we estimate the number of additional cases of gonorrhea that would be treated successfully with first-line antibiotics under this strategy with $S_0 \frac{\Delta L}{L_0}$, where $S_0$ is the number of cases successfully treated with drugs A or B and $L_0$ is the effective life span of drugs A and B under the status quo.

The simulation window of 50 years was selected to ensure enough time for the resistance to emerge against drug A and drug B (in a sensitivity analysis, we set the simulation window at 25 years). We summarized results using the mean and 95% uncertainty interval (i.e., the interval between 2.5th and 97.5th percentiles of realizations) across 500 simulated trajectories. For the "Threshold + trend" and "Enhanced threshold + trend" strategies (Table 1), the 2 thresholds used to inform switching (i.e., threshold for resistance prevalence and the threshold for change in the resistance prevalence) are determined using the optimization algorithm described in S4 of S1 Text.

## Results

We fitted our model against gonorrhea prevalence, the rate of reported gonorrhea cases in 2017, and the proportion of gonorrhea cases with symptoms and estimated the proportion of cases resistant to drugs A, B, or both when 5,000 annual gonorrhea cases are tested for drug resistance during each simulation (Fig 2). We used 5,000 annual cases based on how many *N. gonorrhoeae* isolates were collected and tested through GISP in 2014 (5,093 isolates) [5].

In Fig 3A, we report the tradeoff between increasing the effective life span of antibiotics and reducing the annual incidence of gonorrhea. The origin in this figure represents the status quo, in which switching policies are triggered when greater than 5% of the isolates tested are resistant [8,10]. Increasing this resistance-prevalence threshold for switching to new antibiotic drugs (moving toward the top-right corner of Fig 3A) increases the effective life span of drugs A and B by using the existing drugs for a longer period. Increasing this switching threshold, however, leads to increases in the expected number of annual gonorrhea cases because delaying the switch to a new antibiotic drug lowers the probability of receiving an effective first-line therapy, thereby extending the expected duration of infectiousness while these cases await detection of treatment failure and treatment with effective second-line therapy. The blue curve in Fig 3A has a slope of 15.3 at the origin. This implies that the 5% switch threshold represents a sacrifice of the effective life span of drugs A and B by 1 year to avert an additional 15.3 gonorrhea cases per 100,000 MSM population per year.

Fig 3A also demonstrates that increasing the frequency at which first-line therapy recommendations are revisited could lead to a substantial increase in the effective life span of drugs A and B without increasing the number of gonorrhea cases. Compared to the current policy, the "Threshold—quarterly" strategy could increase the effective life span of drugs A and B by 0.82 years without increasing the number of gonorrhea cases (this is measured as the horizontal distance between the points where the curves in Fig 3A crosses the x-axis). This is equivalent to successfully treating an additional 79.6 (47.4, 111.2) gonorrhea cases per 100,000 MSM population each year with drugs A and B without worsening the burden of gonorrhea.

Fig 3B shows that the "Threshold + trend" strategy, which uses both the resistance prevalence and the change in resistance prevalence since the last year, outperforms the "Threshold —annual" strategy. Compared to the status quo, the "Threshold + trend" strategy could increase the effective life span of drugs A and B by 0.83 years (which is equivalent to successfully treating an additional 80.1 (47.7, 111.9) gonorrhea cases per 100,000 MSM population each year with drugs A and B) without increasing the incidence of gonorrhea. Specifically, the "Threshold + trend" strategy, which removes an antibiotic from the first-line therapy either when the resistance prevalence exceeds 10.1% or when the increase in the resistance

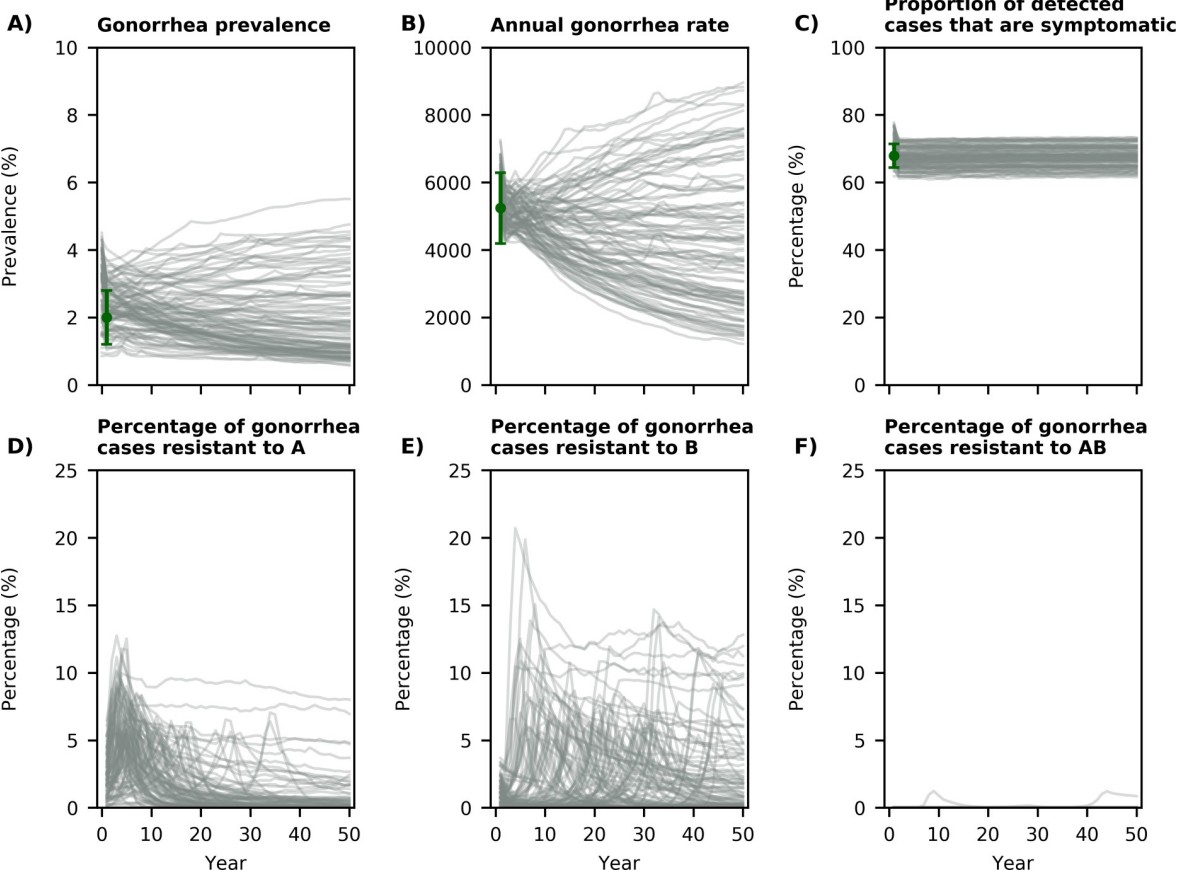

**Fig 2. Displaying 100 simulated trajectories from the calibrated model.** The green dots in panels A–C represent the data or estimates the model is calibrated against: gonorrhea prevalence (2.0% [1.2%, 2.8%] [18,19] of MSM), the estimated rate of gonorrhea cases in 2017 (5,241.8 cases per 100,000 MSM [1]), and the proportion of gonorrhea cases among MSM that are symptomatic (67.9% [64.4%–71.4%] [20]). In these simulated trajectories, the first-line treatment is changed when more than 5% of the annual gonorrhea cases are resistant to the first-line drug. MSM, men who have sex with men.

prevalence from last year is greater than 1.6 percentage points, is expected to increase the effective life of drugs A and B while preventing gonorrhea cases compared with the status quo.

Fig 3C demonstrates that the benefits of the "Threshold + trend" strategy can be enhanced when the evaluation of resistance prevalence is performed quarterly, and the annual number of gonorrhea cases tested for drug susceptibility is doubled. Compared to the current approach, the "Enhanced threshold + trend" strategy could increase the effective life span of drugs A and B by 0.94 years (which is equivalent to successfully treating an additional 91.1 (54.3, 127.3) gonorrhea cases per 100,000 MSM population each year with drugs A and B) without worsening the burden of gonorrhea.

## Discussion

We used a mathematical model of gonorrhea transmission to evaluate how different strategies to inform recommendations for the first-line treatment of gonorrhea would impact the effective life span of antibiotics and the incidence of gonorrhea in the US MSM population. We used a Bayesian approach to calibrate the model to the estimated prevalence of gonorrhea, the rate of gonorrhea cases, and the proportion of cases presenting symptoms among MSM in the US. We examined alternative strategies to inform the timing of shifts in first-line treatment

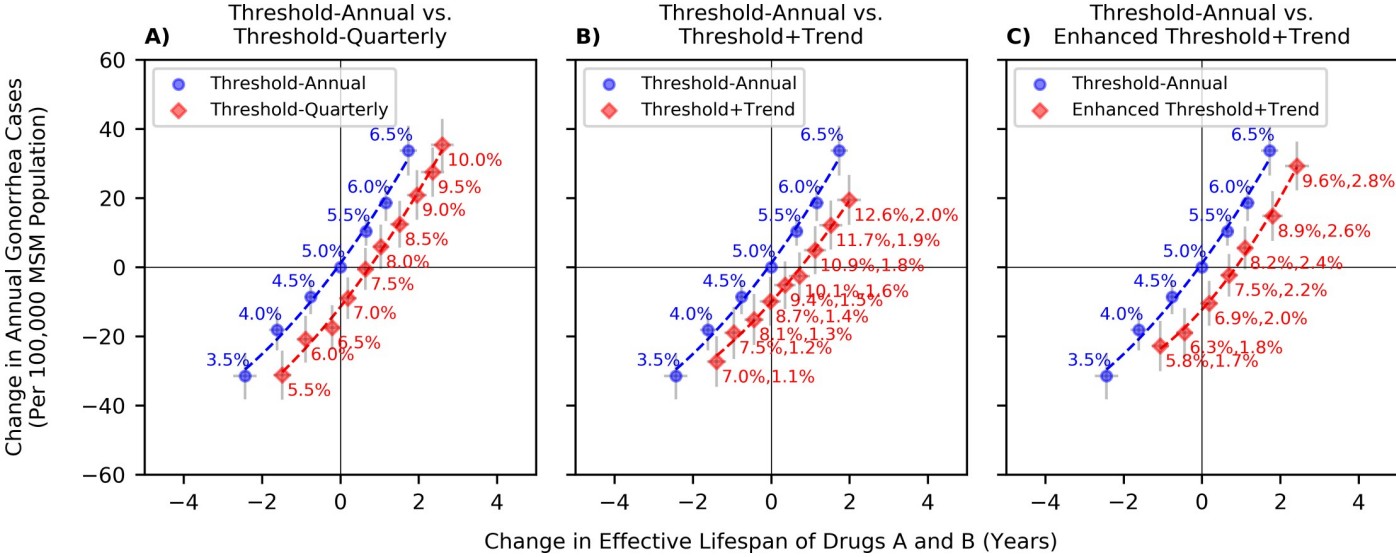

**Fig 3. Comparing the performance of policies in Table 1 with respect to the current policy.** The origins in these figures reflect the current policy that recommends switching the antibiotic used for empiric treatment once the estimated resistance prevalence exceeds 5% [8,10]. The numbers on the curves of "Threshold—annual" and "Threshold—quarterly" strategies represent the threshold of resistance prevalence to switch the first-line therapy of gonorrhea, and the 2 numbers on the curves of "Threshold + trend" and "Enhanced threshold + Trend" strategies represent the 2 thresholds used to inform switching: resistance prevalence (first %) and percentage point change in the resistance prevalence (second %). S6 Fig in S1 Text shows that the comparative performance of these strategies is maintained when the simulation length is reduced from 50 years to 25 years. MSM, men who have sex with men.

regimen. These strategies respond to the data from surveillance systems 1) by revisiting the treatment guidelines more frequently (quarterly versus annually) or 2) by considering not only the current resistance prevalence but also the increase in resistance prevalence since the last decision point to inform the first-line treatment recommendations. Our analysis showed that these adaptive strategies could extend the effective life spans of existing antibiotics for the treatment of gonorrhea without exacerbating the burden of gonorrhea.

In the absence of rapid drug-susceptibility testing to determine the resistance profile of a gonococcal infection, the treatment of gonorrhea remains empiric and based on population surveillance. Historically, once the estimated resistance prevalence for the recommended first-line antibiotic exceeds 5%, it is replaced in the guidelines by a regimen with lower levels of population-wide resistance [8,10]. Our analysis suggests that the optimal choice of this threshold requires a tradeoff between the effective life span of antibiotics and the incidence of gonorrhea. Increasing this switch threshold would increase the effective life span of existing antibiotics but could also increase the burden of gonorrhea; conversely, decreasing this switch threshold would prevent more gonorrhea cases but at the expense of reducing the effective life span of existing antibiotics. Using our mathematical model, we estimated that the 5% switch threshold currently used represents a tradeoff of forgoing a year of the effective life of existing antibiotics to avert an additional 15.3 cases of gonorrhea per year per 100,000 MSM population. Different decision rules could improve this relationship.

Our analysis has a number of limitations. Our mathematical model describes the transmission of *N. gonorrhoeae* only among MSM in the US. The burden of gonorrhea and drug-resistant gonorrhea is particularly high in this subpopulation [1,5], and hence, our conclusions might not be generalizable. For populations with lower burden of the disease, the benefits of adaptive strategies might diminish as the consequences of making suboptimal decisions would be less severe. While data from surveillance systems indicate an upward trend in the rate of gonorrhea cases among MSM [1], we assumed that the incidence and prevalence of gonorrhea

among this population are expected to be relatively stable around the 2017 estimates (Fig 3). We did not model specific antibiotics and instead chose to model hypothetical drugs with characteristics similar to the antibiotics commonly used in treatment of gonorrhea. This allowed us to better capture the uncertainty in the characteristics of current and future antibiotics drugs (e.g., probability of resistance from treatment).

Current US Centers for Disease Control and Prevention (CDC) treatment guidelines for gonorrhea recommend dual therapy with ceftriaxone and azithromycin, but our decision model assumes that first-line therapy consists of only one antibiotic, such as the guidelines now in place in the United Kingdom [21]. Although our approach considers single antibiotic treatment for clarity, it can be extended to scenarios in which combination therapy is the first-line gonorrhea treatment. We assumed that once an antibiotic treatment for gonorrhea is abandoned because of the level of resistance, it will not be reintroduced. However, alternative stewardship and diagnostic strategies (e.g., the use of sequence-based diagnostics to identify the resistance profile [22]) suggest the possibility of reintroduction of these antibiotics; e.g., a recent modeling study suggests that cefixime, which had previously been removed from clinical use because of increasing levels of resistance, could be reintroduced to treat a minority of cases, assuming that cefixime resistance incurs a fixed fitness cost [23].

Our model did not account for site-specific infections, although the percent of infections that are asymptomatic varies by anatomic sites [24–26]. While we assumed that estimates of resistance prevalence calculated from GISP data are representative of the MSM population, GISP includes isolates from the first 25 men (not only MSM) who have been diagnosed with urethral gonorrhea after attending sexually transmitted disease clinics in select US cities. Our model assumes complete adherence to the first-line treatment guidelines. While the actual treatment regimens used in the population may differ from the recommended guidelines, recent studies estimate the adherence to the CDC guideline for the treatment of gonorrhea to be around 80% [27]. Relaxing these assumptions could improve the accuracy of projections made by our model, but it is not expected to significantly affect the comparative evaluation of strategies considered here.

Enhancing surveillance systems to enable more frequent reporting and evaluation of more gonococcal isolates would increase the cost of surveillance. While the cost effectiveness of these proposed changes needs to be studied, the analysis presented here highlights the importance of maintaining effective surveillance systems and the engagement of policy makers to turn surveillance findings into timely decisions to better control the spread of drug-resistant gonorrhea [28]. In the future, decision support tools like the one we proposed in this paper could help policymakers to respond more efficiently to the rise of antibiotic-resistant gonorrhea in a way that could prolong the effective life span of existing antibiotics and control the burden of the disease.

While we await a breakthrough (new antimicrobial agents, novel molecular assays to determine susceptibility to antimicrobial agents, or a gonococcal vaccine), it is important to optimize the use of surveillance systems to minimize the burden of gonorrhea and to slow the spread of antibiotic-resistant strains. We demonstrated the potential for data from surveillance programs to be used in a more efficient and active way to prolong the effective life spans of existing antibiotics without increasing the burden of the disease.

## Supporting information

**S1 Text. Additional details on model structure, calibration procedure, and the algorithm to identify adaptive policies.**
(PDF)

## Acknowledgments

Disclaimer: The findings and conclusions in this article are those of the authors and do not necessarily represent the official position of the Centers for Disease Control and Prevention or the U.S. Department of Health and Human Services.

## Author Contributions

**Conceptualization:** Reza Yaesoubi, Ted Cohen, Katherine Hsu, Thomas L. Gift, Harrell Chesson, Joshua A. Salomon, Yonatan H. Grad.

**Data curation:** Reza Yaesoubi.

**Formal analysis:** Reza Yaesoubi.

**Funding acquisition:** Ted Cohen, Joshua A. Salomon.

**Investigation:** Reza Yaesoubi.

**Methodology:** Reza Yaesoubi.

**Software:** Reza Yaesoubi.

**Supervision:** Ted Cohen, Joshua A. Salomon, Yonatan H. Grad.

**Validation:** Reza Yaesoubi, Katherine Hsu, Thomas L. Gift, Harrell Chesson, Joshua A. Salomon, Yonatan H. Grad.

**Visualization:** Reza Yaesoubi.

**Writing – original draft:** Reza Yaesoubi, Ted Cohen, Yonatan H. Grad.

**Writing – review & editing:** Reza Yaesoubi, Ted Cohen, Katherine Hsu, Thomas L. Gift, Harrell Chesson, Joshua A. Salomon, Yonatan H. Grad.

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
