## [Decision Letter · Decision Letter 0]

27 Sep 2019

Dear Dr. Yaesoubi,

Thank you very much for submitting your manuscript "Adaptive guidelines for the treatment of gonorrhea to increase the effective lifespan of antibiotics: A mathematical modeling study" (PMEDICINE-D-19-02321) for consideration at PLOS Medicine. 

Your paper was evaluated by a senior editor and discussed among all the editors here. It was also discussed with an academic editor with relevant expertise, and sent to three independent reviewers, including a statistical reviewer. The reviews are appended at the bottom of this email and any accompanying reviewer attachments can be seen via the link below:

[LINK]

In light of these reviews, I am afraid that we will not be able to accept the manuscript for publication in the journal in its current form, but we would like to consider a revised version that addresses the reviewers' and editors' comments. Obviously we cannot make any decision about publication until we have seen the revised manuscript and your response, and we plan to seek re-review by one or more of the reviewers. 

We expect to receive your revised manuscript by Oct 18 2019 11:59PM. Please email us (plosmedicine@plos.org) if you have any questions or concerns.

We look forward to receiving your revised manuscript. 

Sincerely,

Thomas McBride, PhD

Senior Editor 

PLOS Medicine

plosmedicine.org

1- In the Abstract Methods and Findings section, please include a bit more information on the population and setting (i.e., the cohort from which you derive the inputs), the specific policy changes investigated, the timeframe simulated, and the main outcome measures.

2- Also in the Abstract Methods and Findings, please quantify the main results (with 95% UIs and p values where relevant).

3- At this stage, we ask that you include a short, non-technical Author Summary of your research to make findings accessible to a wide audience that includes both scientists and non-scientists. The Author Summary should immediately follow the Abstract in your revised manuscript. This text is subject to editorial change and should be distinct from the scientific abstract. Please see our author guidelines for more information: https://journals.plos.org/plosmedicine/s/revising-your-manuscript#loc-author-summary

4- In the first paragraph of the Discussion, please summarize what was done before describing the findings.

5- Please present and organize the Discussion as follows: a short, clear summary of the article's findings; what the study adds to existing research and where and why the results may differ from previous research; strengths and limitations of the study; implications and next steps for research, clinical practice, and/or public policy; one-paragraph conclusion.

Comments from the reviewers:

Reviewer #1: The manuscript by Yaesoubi and colleagues investigates a topical question in gonorrhea research, namely how to design optimal treatment strategies and guidelines. The authors compare a number of alternative strategies to the current practice of switching first-line antibiotics after the level of resistance exceeds 5%. The results suggest that some adaptations of the current surveillance programs would allow to extend the use of antibiotics and/or decrease the burden of disease.

From reading the manuscript, it becomes clear that the authors have clearly identified an interesting problem and propose a thoughtful approach to tackle it using a mathematical model. The modeling framework, a stochastic implementation of a previously published model, appears to be well-adapted.

While I think this is a really nice and important piece of work, the manuscript appears a little incomplete and might benefit from some further investigation. I have a number of suggestions that the authors could consider in a revised version of the manuscript which, I believe, would considerably help strengthen the overall conclusions:

1. Sensitivity analyses: My impression is that the results of the paper might heavily depend on a number of critical assumptions. For example, it is unclear to me why the authors chose a time window of 50 years. Also, increasing the number of additional antibiotics (C, D, ..., or N, K, ...) could result in quite some different results. I think some sensitivity analyses in that respect might be warranted.

2. Uncertainty: The authors write that they summarize results "using the mean and 95% confidence intervals across 500 simulated trajectories". However, I did not find any reporting on confidence intervals or any assessment of uncertainty. Fig. 3 is truly excellent and contains an incredible amount of information, but one wonders how much uncertainty is expected there. Table 1 could also be extended with some key results for the resistance thresholds of 5% and 10%, for example.

3. Methods: I found it a bit difficult to completely follow the modeling procedure. First, it is unclear why the authors chose the simulation approach described in S1.2 instead of the exact stochastic Gillespie algorithm. Did the authors opt for a fixed time step because of computational constraints? Second, it is also not clear how model calibration is performed. Apparently, the authors run 100,000 trajectories but it is not described how the posterior distributions are obtained based on the likelihood. Finally, I could not exactly follow what happens in the threshold-trend strategies described in S4. It appears the authors want to identify tau and theta, but I could not find any reporting of these values (at least not of theta) in the manuscript. Some additional details on this specific scenario and its results would be helpful.

Minor points:

1. Figure 2: It would help if the panels in this figure highlighted the median of all simulations, or an illustrative or typical simulation run.

2. Supplementary Information, p. 5, line 423: "estimated" should be "estimate".

Reviewer #2: The manuscript represents a valuable study into the potential impacts of both switching and optimising criteria for moving between first and second line and second and last line antibiotic therapies.

The study is almost entirely a simulation study. A detailed, yet realistic stochastic epidemic transmission model is established, which has a number of parameters as presented in Table 1. I have one query about the model:

1. In the SI, in Section 1.3, t_0 is defined to be the time at which the increase in the relative infectiousness of resistance profile i should occur. How is this time determined? It's not in the parameter list. Is γ_i(t) a constant before this time?

Model outputs are then related to three data points - two of which are extracted from published estimates and confidence intervals. For these data points an approximate likelihood is drawn up. The analysis of these data points is claimed to be Bayesian.

2. The likelihood is defined, but it is unclear how exactly it is used to produce the posterior distributions. The paper says that N = 100,000 epidemic trajectories are simulated using parameters sampled from the prior distributions. Some trajectories are not considered if they don't satisfy various feasibility constraints. On the remaining trajectories, what happens? Each will have an associated likelihood value, but it is not clear how these are used to produce the posterior distributions? Is MCMC used somehow? Are only a further proportion of the trajectories retained on the basis of their likelihood? This really isn't clear and left me rather confused. It looks like only four parameters have posterior distributions that are drastically different from their prior - which makes sense given there are only three data points. I think there has been a typo in the range of the priors and posteriors for the initial gonorrhea prevalence, as they are not consistent with each other. Also, minor point, but for parameters such as the probability of drug resistance, the prior specified is uniform, but it seems that the order of magnitude is uncertain. The current uniform prior places very little prior probability on this being 10^-5 or 10^-6, which is not the intention, I'm sure. It would make more sense to place the prior on the log-scale.

3. Does removing trajectories that never give a prevalence of resistance of 5% bias against strategies that use a thresholding rule <5%? Are the three criteria in lines 440-442 really so infeasible over the next 50 years?

4. There are many ways to calculate a confidence interval on the basis of binomial data. However, I don't think it is reasonable to be anticipating that the width of such an interval is based on the quantiles of a t-distribution. t-tests are used in cases where the variance of the data is unknown and has to be estimated separately to the mean. However, the estimate for variance of binomial data follows once the probability parameter has been estimated. Therefore, a z-statistic would be used. Can the authors justify their use of the t-distribution rather than the z? Given the likely sample sizes being estimated, this is very unlikely to make any real difference to the analyses, though does simplify the calculations. There's also a typo in the SI on line 420, it should be K hat, not S hat that is being estimated.

Of the retained trajectories, the impact of the four considered policies are assessed, with a detailed description of how each of the policies are optimised.

5. The Results section discusses some improvements that could be made given certain thresholds. It would be good to state explicitly (perhaps I missed it) if these thresholds are the ones found through the optimisation process outlined in the SI?

6. Have any values of ω been used previously? Also, there is no discussion of the anticipated costs of the enhanced surveillance ... are the improvements in the extension of the useful life of the antibiotics cost-efficient?

7. The final paragraph of the SI discusses ω_1, ω_2, ω_3 being drawn from the minimum, origin and maximum values from Fig. 3. I'm right in thinking these are not conventional minima and maxima, rather that they just come from the gradient at the largest and smallest values of the threshold that were considered?

8. Minor point, there are some typos in formulae:

 - should there be a '-' in Equation (6) of the SI, rather than a '+'? 

 - in line 484, p_n approaches zero at a slower rate than p_n? I think the first should be ε_n.

 - the equation in line 356 is confusing, if j is the index on the LHS, it shouldn't be used in the summations, which should be switched to k or some other letter.

 - re-use of notation β in Section 3.3 of the SI. This is also used for the transmissibility. Could a different symbol be used?

Reviewer #3: This manuscript presents a transmission modelling study of innovative drug cycling strategies for the treatment of gonorrhea in the US MSM population. The ideas are intriguing and the ideas described here are important.

Unfortunately, I found it slightly too theoretical for PLOS Medicine.

There are only a few drugs used routinely around the world for treating this pathogen and their rates of acquiring resistance and their initial drop in fitness has been quantified. In a piece aimed at influencing policy, given these other findings are out there, it doesn't seem reasonable to drop back to a purely theoretical Drug A versus Drug B versus Drug N. The direction of effects of more rapid decisions or higher thresholds are reasonably intuitive (but still need pointing out). Readers of this style of article at PM will be looking for robust evidence that the strength of the effect has been accurately assessed - or if not accurately, at least as accurate as is possible with current data and approaches. 

Although much of the language was clear and the charts were well designed. There were key aspects of the reporting that I couldn't appreciate even on careful reading. See comments about my ability to interpret magnitude for both the x and y axis of figure 3. I suspect that the value of this figure is higher than I was able to see, but it was quite opaque to me.

Given the call out to the UK policy in the discussion, it seemed strange to me that the authors were not influenced more by doi.org/10.1371/journal.pmed.1002416 in this journal. I don't know for sure, but I think this prior paper directly influenced the decision they mention.

Just to reiterate - there is genuine innovation here and a whole set of possible strategies not looked at before in the literature. But this presentation of the ideas did not seem to be specific enough to justify the strong statements to policy-making readers of the journal. 

Detailed comments:

Line 34; The caveas in the results section seemed a little strange. Maybe in the conclusions instead? 

42; Seems unusual not to have mentioned the modelling study by Whittles et al that appeared in this journal. They quantified relative fitness of different strains using a model, which seems highly relevant here.

99; Maybe call this threshold-annual

130; How much less transmissible. That seems like a key assumption that needs some discussion. Ref 15 doesn't seem to have any specific estimates of relative transmissibility

133; How exactly is this done and what are the approximate ranges for initial fitness cost and eventual costs. 

141; A table of key parameter assumptions, prior distributions and posterior estimates would be very helpful here

148; please state the justification for the lifespan definition. It doesn't make sense to me. $T$ appears in the bounds of the summation and, I think, in the term itself. Couldn't understand this.

Fig 3: See comments about the lifespan definition. I couldn't understand what this was telling me at this point. Is the y axis absolute or some kind of relative percentage. If absolute, its tricky to interpret because I have to look back at the first figure.

168: So Fig 3 is per 100,000. COmparing this with Fug 2b, they seem to be very modest changes?

180; maybe a discussion of the accuracy with which the trend would need to be known is important. If not here then later. Accurately measuring small changes would not be wihtout costs.

202; this paragraph is a simple reststement of results. It doesn't quite make sense here in the discussion.

217; I'm pretty sure that the UK policy was influenced by the work by whittles et al in this journal that inferred fitness costs for specific drugs.

[LINK]

---

## [Decision Letter · Decision Letter 1]

4 Feb 2020

Dear Dr. Yaesoubi,

Thank you very much for re-submitting your manuscript "Adaptive guidelines for the treatment of gonorrhea to increase the effective lifespan of antibiotics: A mathematical modeling study" (PMEDICINE-D-19-02321R1) for review by PLOS Medicine.

I have discussed the paper with my colleagues and the academic editor and it was also seen again by 2 of the previous reviewers. I am pleased to say that provided the remaining editorial and production issues are dealt with we are planning to accept the paper for publication in the journal.

[LINK]

We look forward to receiving the revised manuscript by Feb 11 2020 11:59PM. 

Sincerely,

Thomas McBride, PhD

Senior Editor 

PLOS Medicine

plosmedicine.org

Requests from Editors:

Data link – the second one you provide is a broken link – please provide working URLs. 

Please avoid the list in the abstract (remove 1 and 2)

Title – Please add the country setting and should MSM be added? 

The abstract seems quite light on quantitative details – please add. 

I think the manuscript would benefit from the description about the "threshold and trend" approach being in slightly more detail--as at line 224-5 (i.e., that the first- to second-line switch occurs at 10.1% prevalence of resistance, or when the prevalence increases by 1.6 percentage points year on year). 

Comments from Reviewers:

Reviewer #1: I would like to thank the authors for adequately responding to my comments on their manuscript.

Reviewer #2: My main concern from the initial review was that the inferential processes were not clear. The authors have now added a significant amount of detail, making things much more explicit and I thank them for this.

The inferential procedure would appear to run according to:

1. Sample 100,000 parameter sets.

2. The likelihood of the three data points is dependent on initial conditions, therefore the likelihood is directly related to the parameters and not dependent on the projected epidemic trajectories... I think...

3. Each prior sample is given a weight according to its associated likelihood, and 500 parameter sets are resampled using weights proportional to the likelihood.

4. The retained 500 samples (and an associated trajectory) are then used as the basis for the assessments of the different antibiotic switching thresholds and surveillance schemes.

I feel a little uneasy about this as prior distributions are notoriously poor importance distributions to sample from. However, as there are only three data points, and most parameters have near identical priors and posteriors, this shouldn't matter unduly in this case. My second concern is a hope that when calculating the trajectories under different switching and surveillance policies, the same random seed is being used so that trajectories match until a particular threshold is reached. Apologies if this is made clear in the text and I have simply missed it.

Some further concerns amendments that are new or remain outstanding (where I'm referring to an item in the supplementary information, I will put an 'S' in front of the line number):

1. Line S437, I'm certain there shouldn't be a \\Delta t in the denominator.

2. Figure S1: Should the lines labelled 'Tx M' at the bottom of the figure be in green? A related point is the equation for S(t) in lines S443-8 - shouldn't this include contributions from those recovering from infection?

3. Section S2, line S467. There appears to be a mixture of \\hat y_t and the p_t that is introduced in the text. Can this be clarified?

4. Line S517: Are we talking about new or prevalent infections?

5. Each of the expressions for L_1, L_2, and L_3 (S484, S497, S509). Why do these have sums from 1 to 10? Are these general expressions for the first ten time periods? If so, then surely both \\hat s and \\hat S should also be indexed by t? I would suggest replacing the 10 with a general T and adding the index, .. or be explicit that you only have data at time t=1.

6. The z statistic has been switched in in Equation S502, but not in S486. Why? I think you need to redo using the z-statistic.

7. Figure 2 - there is clearly a trajectory that has prevalence >5%. Why has this one not been removed?

8. Lines 184: The use of S and L, it's a minor pedantic point, but could you use S_0 and L_0, to make it clear that these are values taken from a baseline analysis?

9. Lines S547, there is an inconsistency regarding the relationship between omega and the switching thresholds. I think you mean to say that lower switch thresholds correspond to higher \\omega.

10. Typos in caption to figure S4. 'Chaning' and 'threshod'.

11. Typo in lines S564-565. 'Values depend' or 'value depends'.

12. S538-540 - can we be clear whether q(\\tau,\\theta) and \\nu(\\tau.\\theta) are values unique to each trajectory or expected values, and whether this expectation is taken wrt to parameter uncertainty or just sampling uncertainty. In the many figures showing the impacts of different thresholds there are uncertainty intervals attached, I presume from looking at the 500 posterior samples for both parameters and trajectories.

13. Typo line S583 'optimzes'.

14. Step 5 of algorithm S2: f_n is not a realisation of f(x), it is an unbiased approximation of it.

[LINK]

---

## [Editor Report · Decision Letter 2]

2 Mar 2020

Dear Dr. Yaesoubi, 

On behalf of my colleagues and the academic editor, Dr. Nicola Low, I am delighted to inform you that your manuscript entitled "Adaptive guidelines for the treatment of gonorrhea to increase the effective lifespan of antibiotics among men who have sex with men in the United States: A mathematical modeling study" (PMEDICINE-D-19-02321R2) has been accepted for publication in PLOS Medicine. 

PRODUCTION PROCESS

PRESS

PROFILE INFORMATION

Thank you again for submitting the manuscript to PLOS Medicine. We look forward to publishing it. 

Best wishes, 

Richard Turner, PhD

Senior Editor 

PLOS Medicine

plosmedicine.org